# A Comprehensive Review of the Development and Therapeutic Use of Antivirals in Flavivirus Infection

**DOI:** 10.3390/v17010074

**Published:** 2025-01-08

**Authors:** Aarti Tripathi, Shailendra Chauhan, Renu Khasa

**Affiliations:** 1Department of Pathology, University of Texas Medical Branch, Galveston, TX 77555, USA; aatripat@utmb.edu; 2Galveston National Laboratory, Galveston, TX 77555, USA; 3Department of Microbiology and Immunology, Miller School of Medicine, University of Miami/UHealth, Miami, FL 33136, USA

**Keywords:** antivirals, flavivirus, entry inhibitors, dengue virus, Zika virus, West Nile virus, Japanese encephalitis virus, yellow fever virus, non-structural protein inhibitors

## Abstract

Flaviviruses are a diverse group of viruses primarily transmitted through hematophagous insects like mosquitoes and ticks. Significant expansion in the geographic range, prevalence, and vectors of flavivirus over the last 50 years has led to a dramatic increase in infections that can manifest as hemorrhagic fever or encephalitis, leading to prolonged morbidity and mortality. Millions of infections every year pose a serious threat to worldwide public health, encouraging scientists to develop a better understanding of the pathophysiology and immune evasion mechanisms of these viruses for vaccine development and antiviral therapy. Extensive research has been conducted in developing effective antivirals for flavivirus. Various approaches have been extensively utilized in clinical trials for antiviral development, targeting virus entry, replication, polyprotein synthesis and processing, and egress pathways exploiting virus as well as host proteins. However, to date, no licensed antiviral drug exists to treat the diseases caused by these viruses. Understanding the mechanisms of host–pathogen interaction, host immunity, viral immune evasion, and disease pathogenesis is highly warranted to foster the development of antivirals. This review provides an extensively detailed summary of the most recent advances in the development of antiviral drugs to combat diseases.

## 1. Introduction

Survival of the fittest is a law of evolution that applies to all entities, including viruses. Virus infection does not always translate to disease. The disease manifestation depends on persistent viral multiplication and dysregulated host immune response. The development of antiviral compounds against flaviviruses is a critical area of research due to the significant global health burden posed by these viruses, which include dengue virus (DENV), Zika virus (ZIKV), West Nile virus (WNV), Japanese encephalitis virus (JEV), and yellow fever virus (YFV). The flavivirus family has RNA as genetic material amplified by RNA-dependent RNA polymerase (RdRp). Owing to the absence of proof-reading ability, the viral progeny are prone to accumulation of mutations, producing a defective, nonfunctional virus or mutated variants with survival fitness, allowing the virus to hijack the host cellular machinery, thus evading the host immune response. This implies that the viral structural and non-structural proteins play a vital role in virus replication, assembly, and egress that can be targeted to develop antiviral therapeutics to curb the infection cycle. Significant progress has been made in developing antivirals against flavivirus, with various strategies being explored. However, the lack of specific therapies remains a challenge, necessitating continued research and innovation in this field. In this review article, we reveal viral and host targets indispensable to flavivirus biology and highlight the current landscape of antiviral strategies against flaviviruses, focusing on key compounds, mechanisms, and research findings. This information would help in understanding the development of novel and antiviral drugs.

## 2. Antivirals Against Flavivirus

Several compounds have been explored to develop antivirals against flavivirus, which will be discussed here extensively. These antivirals are categorized based on different stages of the virus life cycle including entry, replication, polyprotein synthesis and processing, assembly, and virus egress (Figure 1). We also included anti-flavivirus drugs with unidentified targets, artificial microRNAs, and exosomes identified as antivirals (Figure 2).

### 2.1. Entry Inhibitors

Flaviviruses are enveloped viruses that employ the viral E glycoprotein to enter cells. Flaviviruses accomplish this through binding with attachment factors on cell membranes, such as glycosaminoglycans (GAGs), integrins, and C-type lectin receptors. The low affinity and non-specific nature of this contact, mediated by domain III of envelope proteins (EDIII), ensures adherence of a large number of virions to the cell surface [1]. Receptor-specific binding and clathrin-coated pockets determine the development of endocytic vesicles and the invagination of the plasma membrane. The virus undergoes conformational changes in the endosome, such as the trimerization of the E protein, which is required for membrane fusion [2]. Following capsid protein rearrangements, viral RNA dissociates and releases into the cytoplasm [3,4]. To intervene in the virus entry process, a plethora of viral-E-glycoprotein-targeting drugs were developed, a couple of which are reviewed in detail in the next section (Table 1).

Cell-based high-throughput screening (HTS) identified BP34610, a small-molecule inhibitor that recognizes the N-terminal stem region of the E protein. The compound (EC_50_ = 0.48 ± 0.06 μM) inhibited replication of DENV and JEV [28], whereas compound 3-110-224.1.1, a synthetic peptide, inhibited DENV (IC_90_ = 0.77 μM), ZIKV (IC_90_ of 4.0 ± 0.6 μM), WNV, and JEV (IC_90_ = 32 μM) by interacting with the envelope protein (E) [20]. Aside from this, computational methods have been used to synthesize peptides targeting protein interfaces involved in viral membrane fusion. This advancement has allowed peptide drugs specifically against the epitope to be designed, with good physiochemical properties and oral bioavailability [44]. In the following sections, we highlight a few crucial inhibitors against flaviviruses.

Z2, a synthetic peptide, is directed against the E protein that is conserved in all flaviviruses. This peptide inhibited infection by ZIKV (IC_50_ = 1.75 ± 0.13 μM), DENV (IC_50_ = 4 and 5 μM), and YFV (IC_50_ = 5 μM) in in vitro assays. Furthermore, Z2 treatment prolonged the survival of ZIKV-infected AG129 mice and reduced the viral load in pregnant ZIKV-infected mice and their developing fetuses. Also, the pups did not exhibit congenital brain development, and the mother did not have a fever or rash, establishing that Z2 is safe for pregnant mice and can be transferred vertically, like ZIKV. The antiviral effectiveness of Z2 can be attributed to its ability to bind with E protein and compromise membrane integrity, hence inactivating virion [5]. Similarly, P5 peptides derived from helix 2 of the JEV E protein stem region demonstrated antiviral efficacy against both JEV (IC_50_ = 3.93 nM) and ZIKV (IC_50_ = 3.27 μM) with 50% inhibitory concentration (IC_50_) in the nanomolar and micromolar range, respectively. P5 peptide acts by blocking the conformational change of E protein induced by acidic pH. Treatment with P5 peptide reduced virus replication and inflammation in the brains of mice infected with JEV. ZIKV-challenged type I and type II interferon-receptor-deficient (AG6) mice had abrogated histological abnormalities in the brain and testes [9]. Moreover, another peptide designated DN59, analogous to the stem domain of DENV, inhibited the replication of both DENV (>99% inhibition of plaque formation at concentrations of <25 microns) and WNV (>99% inhibition at <25 microns) [18,45].

DET2 (IC_50_ above 500 µM) and DET4 (IC_50_ of 35 μM) are synthetic 10-mer peptides analogous to EDIII of DENV-2 that exhibited antiviral activities against DENV infection. Transmission electron microscopic (TEM) images confirmed that these peptides impair structural and conformational changes in viral E glycoproteins [10,11], thereby preventing the entry of DENV into the cell. In addition, the dipeptide termed EF potentially reduced DENV viral genome replication in all DENV serotypes. EF interferes with the process of membrane fusion by targeting the hydrophobic pocket of E proteins [12]. DENV-2 treated with 200 μM of EF showed reductions in the intracellular E protein and the viral genome by 83.47% and 84.15%, respectively. RI57, short-chain peptides with 28 amino acids, demonstrated inhibitory activity against DENV serotypes (IC_50_: DENV-1 = 4.6 ± 3.2 μM, DENV-2 = 7.4 ± 2.8 μM, and DENV-3 = 5.9 ± 4.8 μM) and ZIKV (IC_95_ = 35 μM). It exerts its effect by inhibiting virus fusion, restricting virus entry into the host cell and subsequent infection [42]. Another peptide, Tat-beclin-1, is an autophagy inducer. Tat-beclin-1 showed antiviral activity against a variety of positive-strand RNA viruses. Tat-beclin-1 D-form (20 mg kg^−1^ i.p.) improves the clinical outcome, lowers the mortality of WNV-infected mice, and ameliorates neuropathology [46].

#### 2.1.1. Monoclonal Antibodies (mAbs)

Quick implementation of several mAb treatments emerged with the outbreak of the COVID-19 pandemic. mAb treatment has likely saved thousands of lives by avoiding the progression of early disease to life-threatening disorders that would otherwise necessitate hospitalization [47]. The first anti-DENV mAb, Ab513 (EC_50_ = 200 ng/mL), engineered to target EDIII of DENV-4, efficiently neutralizes all four DENV serotypes [14]. Prophylactic treatment in mouse models infected with DENV-2 substantially lowered viral load and enhanced survival, showing its antiviral effectiveness in the context of heterologous enhancing antibodies. In addition, human monoclonal antibody (HmAb) 2D22 binds to viral envelope (E) proteins in a dimeric form, hindering the E protein rearrangement essential for virus fusion. Similar to Ab513, HmAb 2D22 neutralizes all four DENV serotypes. HMAb 2D22’s epitope could be a vaccination or therapeutic target. Administration of 2D22 (20 µg/200 µL, i.p.) averted death in AG129 mice challenged with a lethal dose of DENV-2, most likely by preventing the antibody-enhanced lethal vascular leakage [14,15,16].

One potential ZIKV medication is human anti-ZIKV immunoglobulin (ZIKV-Ig), which is currently the subject of a phase 1 double-blind, randomized, placebo-controlled trial [17]. Additional therapeutic antibodies, such as TY014, Tyzivumab, and MGAWN1, were created [48] to neutralize WNV, ZIKV, and YFV infections. These three antibodies underwent trial analysis; however, TY014 completed the phase 1 clinical trial successfully, while Tyzivumab was withdrawn from one of two phase 2 clinical studies due to issues with the cohort evaluation caused by an insufficient number of ZIKV-infected patients. Like Tyzivumab, phase 1 data for MGAWN1 showed that this recombinant humanized monoclonal antibody is effectively and safely tolerated in healthy subjects. MGAWN1 had to be dropped, similar to Tyzivumab, due to inadequate enrollment [19].

AC-10 (IC_50_ < 25 ng/mL), neutralizing monoclonal antibodies isolated from a patient with an active ZIKV infection, effectively neutralized virus infection in Vero cells. It also enhanced internalization of virions into human leukemia K562 cells [43]. ZKA190 (15 mg/kg, i.p.) and FIT-1 (15 mg/kg, i.p.) displayed high neutralizing efficacy and prevented morbidity and mortality of ZIKV-infected mice [21]. Further, B cell epitope mapping studies on ZIKV-infected subjects identified ZIKV-117, a human monoclonal antibody. ZIKV-117 showed strong ZIKV neutralization through its binding affinity towards E protein. Treatment of ZIKV-infected pregnant and non-pregnant mice with ZIKV-117 (100 µg, i.p.) abrogated histopathological changes in the placenta and reduced ZIKV infection and mortality in pregnant mice and their fetus [22]. In contrast, two subsets of E-dimer epitope (EDE) antibodies, EDE1 and EDE2, isolated from a dengue patient, potentially neutralize the serogroups of ZIKV (EDE1/EDE2, IC_50_ = 0.1 nM) and DENV (EDE1, IC_50_ = 0.39 nM (DENV-1), 0.24 nM (DENV-2), 0.64 nM (DENV-3), and 1.13 nM (DENV-4); EDE2, IC_50_ = 0.11 nM (DENV-1), 0.07 nM (DENV-2), 0.11 nM (DENV-3), and 7.79 nM (DENV-4)) [23]. Further, mAb WNV-86 potently neutralized WNV with a 50% inhibitory concentration of 2 ng mL^−1^. A single dose of WNV-86 following 2 days of WNV infection protected mice from the lethal WNV challenge [24]. Thus, this implies that a structure-based rationale paves the way for future antibody-based therapeutic approaches. Monoclonal antibody therapy has several disadvantages, such as limited availability and accessibility, high cost, the necessity of early intervention, and the risk of side effects [49]. Particularly with mAbs, it is important to take into account the possibility of antibody-dependent enhancement (ADE) exacerbation, as mAbs against RSV, MERS, and SARS-CoV-2 have been demonstrated to induce ADE [50].

#### 2.1.2. Synthetic and Natural Inhibitors

Synthetic agents with broad-spectrum action against numerous flavivirus species are of particular interest. An array of cyanohydrazones were identified to avert DENV, ZIKV, and JEV infection. Particularly, JBJ-01-162-04 (IC_90_ = 1.5 ± 0.4 μM) suppresses DENV in an in vivo model. Similarly, 2,4-diamino pyrimidines (2-12-2, IC_90_ = 5 μM; 7-148-6, IC_90_ = 25 μM) and 4,6-disubstituted pyrimidines (GNF-2, IC_90_ = 15 μM; 1-100-1, IC_90_ = 7.5 μM) target the DENV E protein’s prefusion form and impede membrane fusion, preventing viral entry [13]. These molecules bind to a βOG pocket between domains I and II of E protein that appears to be conserved among DENV, WNV, ZIKV, and JEV.

Gossypol, a natural substance, effectively prevented ZIKV infection in nearly all 10 strains. It also demonstrated minimal to no cytotoxicity against four serotypes of DENV human strains. Gossypol (ZIKV: IC_50_ = 3.48 µM; DENV-1: IC_50_ = 1.87 µM; DENV-2: IC_50_ = 1.89 µM; DENV-4: IC_50_ = 2.6 µM) was found to bind to the ZIKV E protein, most likely EDIII, suggesting that it could identify regions and segments of the ZIKV and DENV E proteins that are highly conserved [25]. Further, the antiviral properties of curcumin have been extensively investigated. It shows antiviral activity against many viruses including chikungunya (CHIKV) (IC_50_ = 3.89 µM), ZIKV (IC_50_ = 1.90 µM), and DENV (IC_50_ = 11.51 µM). Prior to infection, the virus treated with curcumin exhibited decreased viral replication, indicating that curcumin prevents the attachment of viruses to cell surfaces [26,27].

Naturally occurring substances, geraniin and palmatine, showed anti-flaviviral efficacy. Palmatine, a protoberberine alkaloid, derived from *Coptis chinensis Franch* inhibits WNV (IC_50_ = 3.6 μM), DENV-2 (IC_50_ = 26.4 μM), and YFV (IC_50_ = 7.3 μM). It has been demonstrated that palmatine prevents infection in a variety of cell lines. It exerts its effect by inhibiting NS2B–NS3 protease activity [31]. Thus, this alkaloid could potentially be used to treat flavivirus infections. Further, geraniin prepared from *Nephelium lappaceum* rind inhibited DENV-2 plaque formation, with an IC_50_ of 1.75 μM. Geraniin binds to the EDIII protein, thus interfering with the initial cell–virus contact in vitro and in vivo. It has been demonstrated that natural ingredients such as flavonoids and naturally occurring phenols like myricetin (IC_50_ = 1.3 ± 0.1 μM), quercetin (IC_50_ = 2.4 ± 0.2 μM), luteolin (IC_50_ = 2.7 ± 0.3 μM), isorhamnetin (IC_50_ = 15.5 ± 0.7 μM), apigenin (IC_50_ = 56.3 ± 0.9 μM), and curcumin (IC_50_ = 3.5 ± 0.2 μM) block ZIKV NS2B–NS3. According to molecular docking, these compounds bind to a pocket at the rear of the active site of ZIKV NS2B–NS3 and block it by allosterically influencing its structure–activity attribute [51]. Of note, dialkylated flavanones named chartaceous A-F (1–6) (IC_50_ = 9.0 ± 3.5 μM) isolated from the bark of Cryptocarya chartacea inhibited dengue virus by interrupting NS5 RNA-dependent RNA polymerase (RdRp) activity, serving as potent non-nucleosidic inhibitors of DENV [52].

#### 2.1.3. Host Function Inhibitors

The existing “one drug, one bug” antiviral strategy frequently leads to the evolution of drug resistance and is not readily scalable to meet the unmet clinical need posed by new viruses.

Targeting the host component is an alternative broad-spectrum antiviral approach with the potential to provide a stronger barrier to resistance. This offers the prospect of repurposing safe, previously approved medications that modulate particular host functions (Table 2). The host-targeted strategy is especially appealing for the treatment of new viral infections for that currently have no treatment [36].

Prochlorperazine (PCZ) (EC_50_ of 88 nM) is a phenothiazine derivative that serves as a dopamine D2 receptor (D2R) antagonist and exhibits antiviral effects against DENV-1, DENV-2, and JEV. A study demonstrated that the drugs inhibit DENV entrance by targeting D2-receptor- and clathrin-associated pathways, thus hampering viral binding and entrance [32]. Moreover, Nawa M. et al. evaluated the effect of chlorpromazine on the entry of JEV. Chlorpromazine blocks the clathrin-mediated endocytic pathway in the case of JEV by accumulating AP-2 and clathrin in endosomal compartments, suppressing the development of clathrin-coated pits. Interestingly, the disruption of clathrin inhibited the replication of JEV (at a dose of 10 μg mL^−1^) and WNV (at a dose of 5 μg mL^−1^) [41,53,61].

Sterol regulatory element binding protein (SREBP) regulates cellular lipidomics. SREBP inhibitors, at a dose of 35 μM, including fatostatin, PF-429242, and nordihydroguaiaretic acid (NDGA) and its derivatives, interfered with the SREBP-dependent lipidomic reprogramming and potently suppressed WNV, DENV, and ZIKV infection in vitro [62]. Likewise, Daptomycin (EC_50_ values between 0.1 and 1.0 µM) is a lipopeptide antibiotic that inhibits ZIKV [33,62].

Nanchangmycin (IC_50_ ranges from 0.1 to 0.4 μM) produced by *Streptomyces nanchangensis* effectively inhibits the entry of the ZIKV along with WNV, DENV, and CHIKV [34]. Antiviral activity against WNV, DENV, and ZIKV has been evaluated for FDA-approved anticancer drugs. Sunitinib, a receptor tyrosine kinase antagonist, and erlotinib, an epidermal growth factor tyrosine kinase inhibitor, were investigated in combination against DENV in IFN-α/β and IFN-γ receptor-deficient murine models. Sunitinib (EC_50_ = 1.962 μM) and erlotinib (EC_50_ = 4.07 μM) treatment reduced viremia and delayed the onset of paralysis in DENV-infected AG-B6 mice. Also, the combination therapy demonstrated antiviral efficacy in monocyte-derived dendritic cells (MDDCs) [36]. Further, adenosine monophosphate-activated protein kinase (AMPK) agonists PF-06409577, metformin, and 5-aminoimidazole-4-carboxamide ribonucleotide (AICAR) demonstrated antiviral potentials in WNV (EC_50_ = 8.2 μM), DENV (EC_50_ = 4.6 μM), and ZIKV (EC_50_ = 0.9 μM) infection and inhibited viral replication through the impairment of AMPK phosphorylation [54]. Further, metformin (50 mg/kg/day) treatment enhanced the survival rate of DENV-infected AG129 mice and attenuated severe DENV infection in a cohort study [63,64]. Furthermore, 25-hydroxylcholesterol (25HC) regulates lipid metabolism. It acts by blocking the fusion of the virus and the cell membrane, thus interfering with the internalization of DENV (IC_50_ = 406 nM), YFV (IC_50_ = 526 nM), WNV (IC_50_ = 109 nM), and ZIKV (IC_50_ = 188 nM), thereby serving as a pan-flavivirus inhibitor. 25HC treatment inhibited ZIKV infection in human cortical tissue in vitro and significantly reduced viremia and mortality in ZIKV-infected BALB/c and AG129 mice. It also demonstrated protection in pregnant mice and alleviated microcephaly in the fetal brain. Moreover, 25HC administration attenuated ZIKV infection and associated clinical signs in rhesus monkeys [37]. Further, PF-05175157 (EC_50_ = 2.7 ± 1.3 μM) an inhibitor of coenzyme acetyl, reduced viremia and conferred protection against WNV infection in a mouse model [55]. A specific inhibitor called GW4869 (doses ranging from 2 μM to 10 μM), which targets sphingomyelinase-2, demonstrated a reduction in WNV and ZIKV in human fetal astrocytes [56,57].

Endosome acidification is critical for viral genome release into the cytoplasm. Chloroquine (CQ), a 9-aminoquinoline, and its hydroxy-analog, hydroxychloroquine, with a long history of usage as an antimalarial drug, have been studied for the clinical treatment of viral diseases. Treatment with CQ (regimen of 600 mg on day 2 and 300 mg on day 3) raises the trans-Golgi network and lysosome pH, which interferes with endosomal fusion and virus maturation. CQ potentially inhibited ZIKV infectivity in human neural stem cells and primary amnion cells [33,40,57]. CQ (dose of 12 and 16 μM) reduced the DENV replication in THP-1 and human dendritic cells. Further clinical trials conducted on DENV patients lowered the incidence of dengue hemorrhagic fever but failed to alleviate viremia [58]. Niclosamide targets the NS2B-binding site on NS3, which is conserved among flavivirus NS3 proteases, allowing it to have broad anti-flaviviral efficacy. It is highly efficient against DENV (EC_50_ = 0.55 ± 0.05 μM), WNV (EC_50_ = 0.54 ± 0.17 μM), YFV (EC_50_ = 0.84 ± 0.02 μM), ZIKV (EC_50_ = 0.48 ± 0.06 μM), and JEV (EC_50_ = 1.02 ± 0.08 μM) [65]. Niclosamide (EC_50_ = 0.37 μM) acts on multiple targets including endosomal deacidification and disruption of the NS2B–NS3 complex, thereby inhibiting ZIKV replication in human neural progenitor cells and animal models. The observed antiviral activity is likely due to its combinatorial effect on both viral and host components [59].

## 3. Viral Replication and Polyprotein Synthesis and Processing

The flavivirus genome is translated into a single polyprotein via the host’s ribosomal machinery. Viral and cellular proteases cleave the polyprotein and yield structural and non-structural proteins. The structural proteins capsid (C), precursor membrane (prM/M), and envelope (E) are integrated into the virion, whereas the non-structural (NS) proteins NS1, NS2A, NS2B, NS3, NS4A, NS4B, and NS5 serve to coordinate the intracellular virus replication, assembly, and modulation of host defense mechanisms [66,67], rendering them promising candidates for the development of therapeutic regimens. Some of the most notable are emphasized in the following subsection (Table 3).

### 3.1. NS1 Inhibitors

NS1 is a non-structural viral protein that facilitates immune evasion, virion assembly, the viral replication cycle, and viral pathogenicity [75,143,144]. It can be found in oligomeric form, co-localized with dsRNA at the site of viral replication. Furthermore, the soluble form of NS1 activates complement-mediated immune suppression and remodels membrane lipids for organizing the replication complex as it engages in the replication of the virus as well. The glycosylation pattern of NS1 has been reported to be conserved at Asn 130 and Asn 207 among all serotypes of DENV, JEV, and YFV [145,146]. This implies that NS1 is a suitable target for the inhibitor development. Celgosivir, a natural alkaloid castanospermine extracted from the *Castanospermum australae*, is an α-glucosidase inhibitor. Celgosivir inhibited DENV infection (DENV-1 (EC_50_ = 0.65 + 0.16 μM), DENV-2 (EC_50_ = 0.22 + 0.01 μM), DENV-3 (EC_50_ = 0.68 + 0.02 μM), and DENV-4 (EC_50_ = 0.31 + 0.12 μM)), mitigated viremia, and prolonged survival by inhibiting glycosylation of prM and E in AG129 mice challenged with DENV [147]. This emphasizes that the glycosylation of NS1 is critical for the virulence of the virus.

The phage display peptide library combined with cutting-edge biopanning techniques identified peptide inhibitors against flavivirus infection. In silico analysis and surface plasmon resonance identified four peptides, i.e., peptide 3, 4, 10, and 11, at a concentration of 10 µM that exhibited strong peptide–protein interactions towards DENV NS1. These peptides differentially inhibited DENV replication in all DENV serotypes in infected Huh-7 cells [68]. Further, using phage display technology, mAbs 3C7 and 4D1 were identified, wherein mAb 3C7 neutralized WNV, while mAb 4D1 neutralized both WNV and JEV [69].

In silico molecular docking screening identified a high affinity of deoxycalyxin-A, a flavonoid, towards ZIKV NS1 [70]. However, in vitro and in vivo research is required to confirm this observation. In addition, α-glycosidase inhibitors like castanospermine (500 μM), deoxynojirimycin (500 μM), and N-nonyl-deoxynojirimycin (20-mg/kg/day, i.p.) decreased NS1 expression and viral production in cells [73,74].

### 3.2. NS2A Inhibitors

Flavivirus NS2A, a hydrophobic protein, is associated with membranes and has a role in RNA replication. NS2A exhibits significant selectivity in binding to the 3′ untranslated region (UTR) of viral RNA and other replication complex constituents. Moreover, NS2A affects the assembly and secretion of virus particles as well as the host antiviral interferon response [148]. So far, no drugs that target the flaviviral protein NS2A have been studied. However, more work needs to be conducted to understand how NS2 interacts with lipids and evades host immune regulation.

### 3.3. NS2B–NS3 Inhibitors

NS3 is a highly conserved protein; its structure comprises two domains: a helicase with an NTPase domain at the C-terminus that aids in the replication of viral genome RNA, and a protease with a trypsin-like serine domain at the N-terminus for polyprotein cleavage. It is noteworthy that viral NS2B is a cofactor for NS3’s protease activity. The coupling of NS3 and NS2B forms a complex that is indispensable for polyprotein processing. Given that NS2B–NS3 plays a crucial role in viral replication, it may be a potential target for antiviral drug studies. A plethora of studies conducted in vitro have demonstrated NS2B–NS3-specific inhibitors including peptidomimetic inhibitors, niclosamide (aforementioned), novobiocin (DENV: IC_50_ = 20,379 nM; ZIKV: IC_50_ = 17,683 nM), and temoporfin (IC_50_ = 1.1 μM). Interaction of novobiocin with the ZIKV NS2B–NS3 binding pocket strongly inhibits ZIKV and DENV replication in cell cultures. It also reduced the viral load and enhanced survival in the treated cohort in mouse models [77,78].

HTS has significantly enhanced pharmacological research through swift and effective selection of novel compounds. This approach led to the identification of temoporfin as an NS2B–NS3 inhibitor. Structural docking revealed that temoporfin binds to NS3 pockets that include key NS2B residues, limiting flavivirus polyprotein processing in a non-competitive way. Temoporfin not only suppressed DENV, YFV, WNV, and JEV in vitro but also reduced ZIKV-induced viremia and mortality in mouse models in addition to inhibiting ZIKV replication in human placental and brain progenitor cells [65].

Split luciferase complementation screening identified JMX0207, a niclosamide that intervenes in NS2B–NS3 interactions. JMX0207 effectively inhibited DENV and ZIKV RNA synthesis in A549 cells and human induced pluripotent stem cells, respectively. Also, JMX0207 (40 mg/kg, oral) treatment diminished ZIKV infection in the 3D organoid and A129 ZIKV mouse model [79]. Another example of drug repurposing is Nelfinavir, an FDA-approved drug. It acts as an NS2B–NS3 and NSP2 protease inhibitor and potentially inhibits DENV-2 (EC_50_ = 3.5 ± 0.4 µM) and CHKV (EC_50_ = 14 ± 1 µM) infection in virus cell-based assays [80].

Similarly, carbazole amidines compound 17, a bis-amidine derivative of N–Me carbazole (IC_50_ = 0.52 μM), showed an antagonistic effect against ZIKV NS2B–NS3 protease. Thus, it efficiently suppressed ZIKV expression in vitro [81]. Further, the non-peptide molecules C30H25NO5 (IC_50_ = 14.9 ± 2.9 μM) and C34H23NO7S2 (IC_50_ > 11.8 ± 0.2 μM) significantly blocked the DENV NS2B–NS3 protease exhibiting in vitro antiviral activity [82]. Virtual and biochemical screening of NCI library compounds identified NSC135618 (IC_50_ = 1.8 μM), an allosteric inhibitor that blocked the conformational change of the NS2B–NS3 protease complex and showed antiviral efficacy against all the flaviviruses including DENV, WNV, ZIKV, and YFV in cell culture [83].

HTS assay employed to identify inhibitors of DENV replication revealed that benzoxazole inhibitors such as ST-610 inhibited the DENV-2 viral titer with an EC_50_ of 0.272 μM and an EC_90_ value of 3.59 μM. ST-610 inhibits double-stranded RNA’s (dsRNA) unwinding activity via acting on the NS3 helicase and preventing it from attaching to viral RNA. Further, ST-610 can reduce viremia in AG129 mice infected with DENV [97].

Likewise, DENV ATPase HTS assay screening discovered ML283 (IC_50_ = 500 nM), a benzothiazole oligomer, that prevented NS3h-catalyzed ATP hydrolysis, thus blocking the RNA unwinding process of WNV and DENV NS3 in cell-based assays [99]. Further, SYC-1307 showed antiviral efficacy against multiple flaviviruses (DENV-2 IC_50_ 0.59 μM, DENV-3 IC_50_ 0.52 μM, WNV IC_50_ 0.78 μM, and ZIKV IC_50_ 0.2 μM), suggesting SYC-1307 is a pan-flavivirus protease inhibitor that targets the hydrophobic pocket of NS3, catalytically inactivating the conformation, which as a result significantly improved the survival of ZIKV-infected mice [98].

Another study by Aleshin et al. provides evidence for the differential substrate selectivity of DENV and WNV proteases. According to structural analysis, aprotinin, also referred to as bovine pancreatic trypsin inhibitor (BPTI), binds to the active site of NS2B–NS3pro and forms an inactive conformation of the oxyanion hole, through an induced-fit mechanism. This complex oxyanion hole is a catalytically competent and stable conformation that perturbs flaviviral proteases in vitro [86]. Further, screening of antimicrobial compounds and in silico docking analysis identified the *benzimidazole compound MB21* as a potential inhibitor of NS2B–NS3 of DENV. The benzimidazole moiety of MB21 (IC_50_  =  5.95 μM) interacts with an allosteric site (Ala125) on the protease that potentially inhibits replication of all the DENV serotypes in vitro with an IC_50_ of 5.9 μM [85].

The screening of a natural compound library and octet binding assay using biotinylated ZIKV pro showed the potential for ZP10 (theaflavin-3,3′-digallate) as ZIKV pro inhibitors. ZP10 (IC_50_ = 2.3 μM) mitigated the viral RNA copy numbers and expression of ZIKV envelope proteins in vitro. The antiviral efficacy of ZP10 has been ascribed to the inhibition of ZIKV polyprotein cleavage by specifically targeting the ZIKV protease [84].

Screening of allosteric exosite-targeting inhibitors coupled with structural modeling revealed NSC157058 as a homolog of NS2B–NS3pro that likely intervenes with the binding of the NS2B cofactor relative to the NS3pro active site, thereby inactivating the protease activity. Incubation of ZIKV-infected human fetal neural and SJL mice with NSC157058 (30 mg/kg) reduced ZIKV infection and viremia [101]. Further, the same strategy indicated that hydroxychloroquine (HCQ), a chloroquine derivative and FDA-approved drug, binds to NS2B–NS3 protease and fits in the active site pocket as well as demonstrating suppression of the number of foci in ZIKV-infected trophoblasts with HCQ treatment with an inhibition constant (*K*_i_) of 92.34 ± 11.91 μM [87].

Interestingly, based on split luciferase complementation (SLC), an FDA-approved drug, methylene blue (MB) and Erythrosin B were identified and characterized to have an inhibitory effect against the viral NS2B–NS3 protease by limiting polyprotein processing, overall leading to a reduced level of viral RNA synthesis and viral protein expression in cell-based assays. The EC_50_ values for ZIKV and DENV-2 are in the low micromolar and nanomolar range in cell-based antiviral assays. In addition, MB (IC_50_ = 29 μM) and Erythrosin B (EC_50_ = 0.3 μM) treatment conferred protection against ZIKV in 3D mini-brain organoids and prolonged the survival of A129 mice infected with ZIKV [89,90].

An in silico docking library search revealed in vitro anti-helicase activity for ivermectin. Ivermectin attenuated the in vitro replication of different flaviviruses [YFV (IC_50_ = 0.12 ± 0.01), DENV (IC_50_ = 0.50 ± 0.07), and WNV (IC_50_ = 0.35 ± 0.04)] by impairing the dsRNA unwinding activity with IC_50_ values in the sub-micromolar range. However, ivermectin failed to protect AG129 mice infected with ZIKV [92,93]. Similarly, HTS identified anti-NS3 helicase activity of suramin. CPE reduction assays demonstrated anti-ZIKV and anti-DENV activity of suramin with an IC_50_ value in the micromolar range. Suramin treatment in vitro caused reduction in ZIKV and DENV genome replication [100,149]. Further, screening of inhibitors that target the NS2B3 protease resulted in the characterization of bortezomib (EC_50_ = 8 nM), which effectively inhibited ZIKV and DENV replication in vitro. Bortezomib treatment ubiquitinates NS2AB3 protease and perturbs its self-cleavage that substantially impaired virus replication [95]. An integrated anchor-based screening approach for screening of the FDA drug dataset against DENV NS3 protease found that asunaprevir (EC_50_ = 10.4 μM) and telaprevir (EC_50_ = 24.5 μM) efficiently lowered viral plaque count in vitro, confirming strong anti-DENV activity [96].

### 3.4. NS4A and NS4B Inhibitors

ER-derived cytoplasmic vesicular packets (VPs) or convoluted membranes (CMs) act as scaffolds for the viral replication complex (VRC). NS4A is one of the key elements of the VRCs, including other NS proteins and viral dsRNA. The N-terminal of NS4A is composed of an amphipathic α-helix, which modulates endoplasmic reticulum (ER) membrane curvature, while the C-terminal of NS4A has a 2k fragment that serves as a signal peptide for the ER localization of NS4B. NS4A mediates anchoring of VRCs to the perinuclear membrane, allowing efficient viral RNA replication. Flavivirus NS4A functions as a cofactor and regulates the ATPase activity of NS3 helicase. In addition, NS4A contributes to the pathogenesis of flaviviruses by antagonizing IFN-I production and RLR signaling, altering the UPR and autophagy as well as hijacking several cellular signaling pathways, resulting in developmental defects. Hence, it represents a promising antiviral drug target [150,151]. Of note, in a cell-line-based model, functional investigation of SBI-0090799 (IC_50_ = 2.1 μM) and Compound B (IC_50_ = 1.32–4.12 μM) showed that they prevented ER membrane remodeling by inhibiting ZIKV and DENV’s NS4A activity, respectively. Thus, they represent a promising lead candidate for further development as an antiviral [102,109].

NS4B is the largest non-structural protein of flavivirus. It shares functional similarities with NS4A. NS4B recruits the host protein to remodel the membrane topology of ER. NS4B interacts with other viral proteins, including NS1, NS2B, NS3, and NS4A, facilitating the formation of a viral replication complex. It pairs with NS3 and enhances the helicase activity of NS3. Also, the interaction with chaperones promotes optimal polyprotein folding and expression. Moreover, NS4B aids in host immune evasion by RNAi pathway inhibition, autophagy and apoptosis modulation, and type I interferon signaling mitigation [152,153]. Because of all these attributes, NS4B is an appealing target for antiviral research. Research leveraging compounds targeting NS4B activity has discovered quite a few compounds with potent antiviral activity.

In particular, JNJ-A07 and JNJ-1802 prohibited the formation of the NS3–NS4B complex. JNJ-A07 induces a conformational shift in the cytosolic loop of NS4B by blocking the NS4B–NS3 interaction. JNJ-A07 is a pan-serotype DENV antiviral since it can also affect a wide range of DENV isolates at nanomolar to picomolar concentrations. Further, JNJ-A07 (10 mg per kg) treatment reduced viremia in DENV-infected AG129 mice. An analog of JNJ-A07 (JNJ-64281802) demonstrated favorable pharmacokinetic and safety profile, thus was registered for a phase 2 randomized, double-blind, placebo-controlled clinical trial to examine therapeutic potential against DENV infections in humans. However, no publication of the trial’s findings has been made [103].

JNJ-1802, a small-molecule inhibitor belonging to the same chemical series as JNJ-A07, showed antiviral activity in picomolar to low nanomolar concentrations against all DENV serotypes, JEV, WNV, and ZIKV in vitro with mean EC50 values ranging from 0.25 µM to 1.1 µM. JNJ-1802 dosing alleviated the viral RNA load and conferred protection in AG129 mice challenged with DENV serotypes (DENV-1, DENV-3, and DENV-4) as well as in monkeys infected with DENV-2 [154]. JNJ-1802 exhibited excellent safety, tolerability, and pharmacokinetics; thus, it successfully completed the phase 1 clinical trial in healthy human participants (NCT05201794) and in patients with confirmed DENV infection (NCT04906980) [104,155].

Investigation of the structure–activity association found that JMX0254 (100 mg/kg, oral) and BDAA (IC_50_ = 0.17 μM) target NS4 activity and inhibited expression of DENV in AG129 mice and YFV, respectively [107,108]. Another compound, SBI-0090799 (2E)-N-benzyl-3-(4-butoxyphenyl) prop-2-enamide) (IC_50_ = 2.1 μM), obstructs the formation of the membrane replication complex by blocking NS4, which suppressed the replication of several ZIKV strains in a cell-based assay [109].

It has been reported that NITD-688 antagonizes NS4B viral protein. However, the specific mechanism is being studied. Cell-based CPE assay verified that NITD-688 was potent against all DENV serotypes. Nuclear magnetic resonance confirmed the direct interaction of NITD-688 with NS4B, as NITD-688 failed to bind with mutated NS4B at T215A and A222V. NITD-688 (30 mg per kg, oral) alleviated viremia in DENV-infected AG129 mice. Further, it showed good in vivo pharmacokinetic properties and oral bioavailability post-administration in mice, rats, and dogs. Overall, the findings validate NITD-688 as a preclinical candidate for DENV therapy [105].

Screening of the FDA-approved drugs library identified manidipine (at a 10 μM concentration) to be efficient against all flaviviruses including JEV, DENV, ZIKV, and WNV in vitro. It prevented brain damage in JEV-infected rats but did not affect viremia. Manidipine substantially lowered viral RNA synthesis by targeting NS4B, but the mechanism of its antiviral activity remains unclear [106]. Lycorine, an alkaloid, attenuated the viral titers of WNV, DENV, and YFV at a concentration of 1.2 microns and potently inhibited flaviviruses in a cell-based assay. Moreover, AZD0530 and dasatinib exhibit substantial efficacy in inhibiting DENV-2 replication in cell culture. NS4B inhibition is the mediating mechanism for these drugs, although their precise processes are yet to be determined [110].

### 3.5. NS5 Inhibitors

The biggest flavivirus protein, NS5 harbors both methyltransferase (MTase) at the N terminal and RdRp at C terminal, playing a vital role in RNA replication. As a result, it has been researched the most for the development of antiviral drugs against flavivirus [156]. There are several types of antiviral drugs that target NS5, including MTase inhibitors, non-nucleoside inhibitors, and nucleoside inhibitors. Drugs have been screened and repurposed to identify nucleoside analogs that target viral polymerases and exhibit flavivirus-opposing action both in vitro and in vivo [157]. A plethora of nucleoside analogs with high antiviral potency and favorable pharmacokinetics have already received FDA approval for use as antiviral medication to treat HIV infection [158,159,160], HCV [161], and herpesvirus [162].

#### 3.5.1. Nucleoside Analogs

The most appealing drugs are usually nucleoside/nucleotide inhibitors (NIs) as they target conserved RdRp active sites prone to relatively low mutation rates in these regions, allowing broad-spectrum activity and a strong barrier to resistance. Structurally, NIs are remarkably similar, but a few small variations in heterobase moiety, kinds of glycosidic linkages, and substitutions at various sugar positions can greatly impact their pharmacokinetic (PK) and active characteristics. Their antiviral activity stems from the replication-induced premature termination of RNA genome synthesis.

An adenosine analog known as BCX4430 (Galidesivir) shows broad-spectrum antiviral activity against Ebola virus (IC_50_ = 11.8 μM), Marburg virus (IC_50_ = 5.0 μM), YFV (IC_50_ = 14.1 μM), JEV (IC_50_ = 43.6 μM), and DENV-2 (IC_50_ = 32.8 μM) both in cell based assays and in mouse models by inhibiting viral RNA polymerase by non-obligate RNA chain termination [112,113]. Phase 1 trials for this drug are currently underway, but the results are not yet available. Furthermore, NITD008, another adenosine analog, showed an inhibitory effect on the infection of DENV-2 (EC_50_ = 0.64 μM), WNV, YFV, ZIKV (EC_50_ = 241 nM), and hepatitis C virus (HCV) (EC_50_ = 0.11 μM) in animal models and cell lines. Phosphorylated NITD008 acts as a chain terminator and directly inhibits RdRp during the synthesis of viral RNA replication, preventing mortality in DENV- and ZIKV-infected mice [118,163]. Similarly, favipiravir, a pyrazinecarboxamide derivative, showed a good tolerance profile in patients and is currently undergoing clinical studies to treat SARS-CoV-2 [164], Ebola virus [165], and influenza virus. Favipiravir (EC_50_ = 61.88 μM) not only conferred protection in mice from meningitis and encephalitis caused by a lethal dose of YFV and WNV [115], but was also successful against ZIKV, indicated through in vitro study, which suggested an increase in the frequency of mutations in the viral genome, leading to generation of non-infectious defective viral particles and thus declining viral infectivity in subsequent progeny [166]. Favipiravir (EC_50_: from 62 to >500 μM) treatment was well tolerated and demonstrated a statistically significant decline in the ZIKV virus load in the plasma of cynomolgus macaques [116]. Monophosphate prodrug analogs of 2′-deoxy-2′-fluoro-2′-C-methylguanosine are strong inhibitors of the RdRp of the HCV and exhibit anti-DENV properties in a cell-based assay. In an AG129 mouse model, cyclic phosphoramidite, prodrug 17, and compound 17 (100 mg/kg) showed decreases in viremia. Nevertheless, the compound failed to attain the “no observed adverse effect level” (NOAEL) in dogs due to mild pulmonary inflammation and hemorrhage. The drug did not progress to the clinical stage because of the severity noted in the preclinical safety assessment [167].

Prodrug balapiravir (R1479) is a cytidine analog that was used in the clinical trial for the treatment of patients infected with chronic HCV, resulting in an improved outcome for patients [168]. In contrast, balapiravir (3000 mg, oral) administration in DENV patients did not differ from drug and placebo treatment in terms of antiviral response, cytokine profile, or time to fever [169]. Likewise, in vitro and in vivo testing of NITD-008, an adenosine nucleoside analog inhibitor, demonstrated strong antiviral efficacy against ZIKV (EC_50_ = 241 nM), TBEV (EC_50_ = 3.31 μM), and DENV (EC_50_ = 0.64 μM) [118,170]. However, preclinical toxicity, including weight loss, decreased motor activity, bloody feces, irreversible corneal opacities, and blood anomaly, prevented the drug from moving forward with clinical trials [163].

The guanosine nucleotide analog prodrug AT-752 has been shown to inhibit RdRp activity of NS5 protein. AT-752 exhibited strong in vitro action against DENV-2 (EC_50_ = 0.48 μM), DENV-3 (EC_50_ = 0.77 μM), WNV (EC_50_ = 1.41 μM), YFV (EC_50_ = 0.31 μM), ZIKV (EC_50_ = 0.64 μM), and JEV (EC_50_ = 0.21 μM). In addition, an AG129 mouse model of DENV infection and hamsters infected with YFV showed a reduction in viremia, improved health score, and prolonged survival with AT-752 administration. Preclinical toxicology studies in mice, rats, and monkeys have indicated no adverse clinical signs. Thus, AT-752 has successfully finished its phase 1 clinical trial (NCT04722627) and is currently in phase 2. Further, the metabolism of AT-752 yields AT9010, an active triphosphate metabolite that prevails stably in cynomolgus monkey and human plasma. During RNA synthesis, AT-9010 competes with GTP for incorporation, causing chain termination, thus efficiently inhibiting RNA synthesis in vitro and in vivo in a mouse model of DENV infection. The antiviral activity of AT9010 is presently being assessed in a phase 1 experiment [120,121].

In vitro testing of another nucleoside analog, 7DMA (7-deaza-2′-C-methyladenosine), also known as MK-608 (EC_50_ = ranging from 5 to 15 μM), effectively halted RNA synthesis of flaviviruses such as TBEV, ZIKV, WNV, and DENV by chain termination of nascent RNA. In vivo, 7DMA (50 mg/kg/day) treatment reduced viral titer along with the aversion of neuroinflammation and disease progression in ZIKV-infected AG129 as well as WNV-infected BALB/c mice. Furthermore, administration of 7DMA during infection resulted in 100% survival of WNV-infected mice, while treatment after three days of infection, at the peak of viremia, led to a 90% survival rate, confirming its antiviral activity [122,123,171].

The quest for a pan-flaviviridae drug may benefit from the highly conserved structural and functional homology of RdRp enzymes across different flaviviruses [172]. In a similar vein, sofosbuvir is a clinically approved small-molecule inhibitor against hepatitis C infection. Three-dimensional modeling revealed that the drug pairs with the palm and finger region of RNA polymerase, where incoming nucleotides are integrated into a nascent strand of RNA, thus impairing RdRp activity. Sofosbuvir has shown in vitro antiviral activity against ZIKV (IC_50_ = 13.6 μM), DENV (IC_50_ = 8.31 μM), WNV (IC_50_ = 11.1 ± 4.6 μM), and YFV (IC_50_ = 4.8 ± 0.2 μM). Notably, sofosbuvir mitigated ZIKV replication and blocked cell death in neural stem cells (NSCs), macrophages/microglia, and brain organoids. Interestingly, pre-treatment and post-treatment of YFV-infected Swiss and A129^−/−^ mice with sofosbuvir (20 mg/kg/day) abrogated hepatic histopathological lesions, weight loss, viral load in the brain, and mortality. Outcomes of YFV-challenged mice were superior with pre-treatment of sofosbuvir than post-treatment [124,125,126,173,174,175].

Ribavirin was originally prescribed to treat HCV and HBV, much like other nucleoside analogs including AT-752, 7DMA, and sofosbuvir. Functionally it acts as an RNA polymerase inhibitor; alleviates RNA replication of DENV, JEV, and ZIKV; and halts cell apoptosis in vitro. Ribavirin (15 mg, i.p.) administration to ZIKV-infected STAT1-deficient mice exhibited prolonged survival, but mice continued to have low viremia levels until death [130,176] as well as failing to inhibit DENV infection in the AG129 mouse model Also, oral ribavirin administration could not rescue JE patients from mortality [123]. However, rifapentine (IC_50_ = 0.5259 μM), a rifamycin antibiotic, is a potent YFV inhibitor. It interferes with viral attachment and impedes RdRp activity, as shown by binding assays and molecular docking, respectively. Rifapentine increases survival rates, mitigates clinical symptoms, and lowers viral load in both YFV-infected type I interferon receptor knockout A129^−/−^ and wild-type C57BL/6 mice. In fact, it surpasses sofosbuvir, a previously described YFV inhibitor in mice [6]. Further, T-1106, a pyrazine nucleoside analog, showed considerably enhanced survival rates in YFV-infected hamsters. Even low doses of 32 mg/kg body weight per day (120 μmol/kg/day) were effective against the virus, indicating its potential use in the treatment of YFV [133].

#### 3.5.2. Non-Nucleoside Inhibitors

This class of compounds obstructs RdRp conformation change during RNA polymerization by direct interaction with the thumb/palm interface of the RdRp. Given that the X-ray crystallography showed compounds such as *N*-sulfonylanthranilic acid derivative NITD-1 (IC_50_ = 7.2 μM), NITD-2 (IC_50_ = 0.7 μM), NITD-640 (IC_50_ = 0.9 to 13  μM), NITD-434 (IC_50_ = from 6 to 17  μM), and NITD107 (IC_50_ = 113 μM), either pair with the RdRp fingers or that targets the junction between the fingers and palm subdomains exhibits anti-DENV activity against all DENV serotypes in vitro [127,177,178]. Structural activity relationship studies revealed that FDA-approved drugs—efavirenz (IC_50_ = 25.78 μM), tipranavir (IC_50_ = 29.85 μM), and dasabuvir (IC_50_ =  16.12 μM)—halt RNA polymerization by premature termination of RNA synthesis of flaviviruses including WNV, ZIKV, and TBEV in cell culture [129]. Further, lycorine, an alkaloid derived from benzyl phenethylamine, was used both in vitro and in vivo to treat ZIKV infection. The treatment (lycorine—5 mg/kg) lowered the virus replication in the CNS, liver, and serum as well as reduced histopathological lesions in the brain and liver of ZIKV-infected AG6 mice. These mice also showed decreased levels of inflammation. Lycorine directly interacts with finger domains of RdRp that potently inhibit RdRp activity and viral replication [135]. Likewise, molecular docking revealed the binding affinity of TBP (IC_50_ = 94 nM) towards the residues in the catalytically active site of RdRp. This likely explains the inhibition of ZIKV replication in vitro as well as the viral load in the plasma of immunocompetent BABL/c mice infected with ZIKV [137]. Similarly, emetine (IC_50_ = 52.9 nM) blocks NS5 function and accumulates in cellular lysosomes with impaired autophagy and lysosomal function, interrupting cellular trafficking of nascent virion and eventually eliminating viral infection in vitro. Further, emetine treatment significantly decreased NS1 and ZIKV viral load in the serum and liver of infected immunocompromised Ifnar1^−/−^ and immunocompetent SJL mice, indicating anti-ZIKV activity [136].

Notably, NS5 harbors nuclear localization sequences (NLSs) which directly interact with importin β1. This interaction is essential for the translocation of the importin-NLS-containing cargo protein complex through the nuclear pore complex (NPC) into the nucleus [179]. Ivermectin (LD_50_ = 150 μM) has been used as a broad-spectrum inhibitor of importin α/β, which effectively prevented DENV replication in vitro. Pharmacokinetic and pharmacodynamic examination of DENV patients dosed with ivermectin suppressed NS1 expression and alleviated dengue disease severity. Indicating potent antiviral activity of ivermectin towards DENV [131].

Of note, AR-12 (also known as OSU-03012) and its derivatives P12-23 and P12-34a act as inhibitors of the de novo pyrimidine biosynthesis pathway via inhibiting mitochondrial enzyme dihydroorotate dehydrogenase (DHODH) activity, leading to the depletion of uridine monophosphate pool, essentially required for the viral RNA synthesis and protein glycosylation. Thus, they effectively impaired RNA synthesis of DENV (IC_50_ = 20.3 ± 0.2 µM), ZIKV (IC_50_ = 21.4 ± 0.2 µM), and JEV (IC_50_ = 21.4 ± 0.2 µM) in vitro and STAT1-deficient mice infected with DENV [60].

#### 3.5.3. MTase Inhibitors

MTases located at the N termini of NS5 perform dual N7 and 2′-O methyltransferase activities, and guanylyltransferase (GTase) is implicated in RNA cap formation. Extensive research studies into MTase inhibitors revealed that sinefungin, an analog of S-adenosyl-L-methionine (SAM), showed greater affinity for the SAM binding pocket of viral NS5, thus attenuating (IC_50_ = 0.03 mM) N-7 and (IC_50_ =  0.041 mM) 2′-*O* methylation by MTase. This likely explains the inhibited replication of both WNV and DENV [138,139]. Flexible nucleoside analogs known as “fleximers” have shown antiviral effectiveness against a variety of viruses, including the Middle East Respiratory Syndrome coronavirus (MERS-CoV), Ebola virus, and lately, flaviviruses such as ZIKV, DENV, and YFV cell-based systems [141,142]. Since non-structural proteins are substantially conserved in structure and are essential for viral replication, they are vital candidates for therapeutic targeting. As a proof of concept, pan-flavivirus antiviral is conceivable due to the structural homology of indispensable viral non-structural proteins. Consequently, the development of pharmacological inhibitors based on NS-protein-centric research targeting NS4B, NS5, and NS3 efficiently halted virus biogenesis.

## 4. Assembly and Egress Inhibitors

All flaviviruses are enveloped viruses employing lipid remodeling of their hosts to change the membrane’s fluidity to assemble itself and provide energy for the reproduction of their genomes [180,181,182]. In the ER, the virus first assembles as an immature particle. It is then carried to the trans-Golgi network (TGN), where it matures in response to pH. The host protease furin cleaves the pr domain of prM during the maturation phase, eventually releasing a fully matured infectious virus particle from the cell [183].

Glycosylation of viral proteins is integral for the release of enveloped viruses and infectivity. (Glc)_3_(Man)_9_(GlcNAc)_2_ is a 14-residue oligosaccharide, added to specific asparagine residues on the prM, E proteins, and NS1 of the newly synthesized polypeptide in the ER [184,185], followed by the trimming of terminal glucose by ER Glucosidase I on high-mannose carbohydrates to generate monoglucosylated glycoproteins that further associate with ER chaperones like calnexin and calreticulin for proper folding [186]. Given the functional implication of glycosylation in the virus life cycle, inhibitors of α-glucosidase have shown inhibitory potency against viruses. Iminosugars such as castanospermine (CST) [187] and deoxynojirimycin (DNJ) [74] hinder the removal of the terminal glucose residue on N-linked oligosaccharides of DENV envelope glycoproteins and NS1, leading to the formation of misfolded proteins and defective viral progeny in vitro and in vivo. Further, celgosivir, also known as 6-O-butanoyl castanospermine (DENV-1–4, EC_50_ =  0.65 + 0.16 μM, 0.22 + 0.01 μM, 0.68 + 0.02 μM, 0.31 + 0.12 μM), demonstrated 100% protective efficacy against lethal infection in DENV-infected AG129 mice [188]. In response to the promising preclinical pharmacology data, a phase 1b randomized, double-blind, placebo-controlled trial of celgosivir was conducted in patients with DENV. However, celgosivir failed to reduce fever or viremia in patients with DENV infection [189]. In addition, no serious adverse events were reported in the phase 1 study of randomized single oral dose safety of UV-4 hydrochloride (UV-4B) [190]. Other promising iminosugars and their derivatives, such as UV-12 (EC_50_ =  21.71 μM), CM-9-78 (EC_50_ =  0.42 μM), and CM-10-18 (EC_50_ =  0.46 μM), demonstrated anti-DENV activity by inhibiting α-glucosidase in vitro and in vivo, so they merit further investigation for application in therapeutic settings [191,192,193].

Flaviviruses modulate the host fatty acid biosynthetic route to alter the lipid composition for nascent virion biogenesis. Rendering pharmacological manipulation of cellular lipids is a desirable antiviral approach. Host-directed antiviral interventions unveiled acetyl-coenzyme A carboxylase (ACC) 1 and 2 inhibitors, such as PF-05175157 [(WNV, EC_50_ =  2.7 ± 1.3 μM), (ZIKV, EC_50_ < 1.2 μM), (DENV, EC_50_ = 1.0 ± 0.3 μM)], TOFA (5-tetradecyloxy-2-furoic acid) (10 μg/mL), and MEDICA 16 (3,3,14,14-tetramethylhexadecanedioic acid) (50 μg/mL), exhibited potency against various flaviviruses, including WNV, DENV, and ZIKV in vitro. as Additionally, PF-05175157 administration significantly decreased viral load in the plasma, brain, liver, kidneys, and lungs of WNV-challenged mice [55,194].

Unlike other host-directed antivirals, SFV785 treatment (at a concentration of 10 μM) could not inhibit viral RNA synthesis or translation within the ER compartment. However, it perturbed the co-localization of the structural E protein with the replication complexes during assembly, which strongly lowered the yield of infectious virions of DENV and YFV [195,196].

Preclinical evaluation of lovastatin (200 mg/kg, oral) using animal models displayed anti-DENV activity. This could be due to a decrease in the cholesterol-rich microdomains and isoprenylated proteins necessary for virus trafficking, thus limiting the entry of the virus into the cell and viremia in vivo. Yet, it failed in a randomized, double-blind, placebo-controlled trial for DENV patients [197,198]. Similarly, ezetimibe (IC_50_ = 19.15 µM) and atorvastatin (IC_50_ = 24.12 µM) showed significant antiviral effects in AG129 mice infected with DENV-2 by targeting the cholesterol biosynthetic pathway [197].

The assembly of viral glycoprotein and capsid is a well-coordinated and highly regulated process that relies on cellular signals and the environment to initiate conformational or allosteric changes in the assembly process. By interfering with the dynamics of viral maturation, assembly, and egress, it is possible to perturb virus propagation (Table 4).

### C Protein Inhibitors

C protein is the structural protein that interacts with the RNA genome and prM and E proteins before packaging. The aggregation of membrane-associated capsid into the nucleocapsid structure induces the formation of immature virus particles. The mature C is a highly basic protein of 12 kDa that forms homodimers in solution [205], while the residues at the N-terminal region of the capsid are implicated in RNA binding and viral particle formation. Despite being the least conserved of the flavivirus proteins, the capsid’s structural characteristics and charge distribution are both well conserved [206]. Extensive structure–activity relationship (SAR) analysis identified an analog VGTI-A3–03 with anti-DENV-2 potency by promoting the formation and release of noninfectious virions. Further analysis revealed that VGTI-A3–03 (IC_90_ = 25 nM) directly pairs with C dimers in viral particles and induces the formation of malformed DENV virions in vitro [202]. Likewise, structural, biochemical, and virologic approaches demonstrated that ST148 (EC_50_ = 0.016 ± 0.01 μM) through hydrophobic interaction forms a complex with C tetramer, which then incorporates into a virion, inducing the production of progeny with defective nucleocapsid uncoating. ST-148 (50 mg/kg/day) administration mitigated the viremia and viral load in the spleen and liver of DENV-challenged AG129 mice. Hence, it showed promising anti-DENV efficacy against all four serotypes in the cell culture and mouse model [203,204]. C protein binds with viral RNA to regulate encapsidation, as well as bind and sequester short interfering RNAs to protect the flavivirus genome from siRNA pathways. Consequently, it presents a desirable target for anti-flaviviruses [207].

## 5. Anti-Flavivirus Drugs with Unidentified Target

Several drugs have demonstrated antiviral effects against one or more flaviviruses both in vitro and in vivo. For some, though, the course of action is still unclear. These medications include azithromycin (EC_50_ = 2 to 3 μM), nitazoxanide (EC_50_ = 0.12 ± 0.04 μg/μM), amodiaquine (EC_50_ = 1.08 ± 0.09 μM), lanatoside C (IC_50_ = 0.19 μM), bromocriptine (EC_50_ = 0.8–1.6 μM), and hippeastrine hydrobromide (HH) (IC_50_ = 1.95 μM) [208,209,210,211]. As with lanatoside C, its low therapeutical index dose makes it unsuitable, whereas HH was successful in controlling infection in human fetal-like forebrain organoid cultures. In a rat model, ZIKV infection was efficiently controlled with a notable reduction in ZIKV RNA in the brain and ZIKV-induced cellular death. FDA-approved fluoroquinolones enoxacin (EC_50_ = 18.1 μM), difloxacin (EC_50_ = 25.4 μM), and ciprofloxacin (EC_50_ = 56.8 μM) inhibited the replication of the flaviviruses Modoc (MODV), Langat (LGTV), DENV, and ZIKV in HEK-293 cells at micromolar concentrations. ZIKV titer was significantly reduced in mice administered enoxacin, 2.5 times lower than the control [212]. Arbidol has antiviral effects at the micromolar level (EC_50_ values ranging from 10.57 ± 0.74 to 19.16 ± 0.29 µM) in Vero cells infected with tick-borne encephalitis virus, ZIKV, and WNV [213].

NGI-1 (EC_50_ = 0.85 μM) demonstrated pan-flavivirus antiviral activity and blocked viral RNA replication of DENV and ZIKV in multiple disease-relevant cell lines [214]. Trametinib is the safest and is efficacious against all of the viruses, inhibiting the replication of ZIKV (EC_50_ = 3.03 μM) (EC_50_ = 7.91 μM) and YFV 1000-fold, and DENV-2/3 (EC_50_ = 2.46/6.33 μM) nearly 100-fold. This pan-antiviral effect indicates that trametinib could be repurposed for the treatment of flaviviral infection [215]. Quinestrol (ZIKV, IC_50_ = 11.4 μM; DENV-2, IC_50_ = 15.3 μM; WNV-KUNV, IC_50_ = 10.1 μM) and raloxifene hydrochloride (ZIKV, IC_50_ = 7.4 μM; DENV-2, IC_50_ = 9.3 μM; WNV-KUNV, IC_50_ = 7.7 μM) are two estrogen receptor modulators that demonstrated minimal cytotoxicity and effective inhibition of ZIKV, DENV, and WNV (Kunjin strain) infection at low micromolar concentrations [216]. Anisomycin, an alkaloid, inhibited DENV (EC_50_ = 31.3 ± 1.2 nM) and ZIKV (EC_50_ = 15.9 ± 1.1 nM). A mouse model of ZIKV treated with a modest dose of anisomycin showed a significant decrease in morbidity and mortality [217]. Fenretinide (EC_90_ = 2.0 μM), also known as 4-hydroxyphenyl retinamide (4-HPR), was shown to be an inhibitor of DENV in cell culture. In an oral DENV infection model in mice, 4-HPR prevents the steady-state build-up of viral genomic RNA and lowers viremia [218]. Furthermore, the antimalarial medication amodiaquine (IC_50_ = 2.28 μM) conferred protection in hNPCs and a SCID-beige mouse challenged with ZIKV [219].

## 6. Artificial microRNAs

The discovery of artificial microRNAs (amiRNAs) directed against viral genomes and the antiviral effect of human miRNAs has abrogated neurovirulence and flavivirus infection (Table 5) [220,221,222]. amiRNAs targeting viral conserved regions at 3′UTR were a useful approach for improvements of nucleic acid inhibitors against JEV. Transient expression of two amiRNAs (amiRNA #1 and amiRNA #2) showed a reduction in intracellular viral RNA and non-structural 1 (NS1) protein and decreased the release of infectious viral particles up to 95% quantified by viral plaque assay [223]. In a similar vein, amiRNA targeting a conserved region of gene NS5 significantly reduced the WNV replication as determined by plaque assay, real-time RT-PCR, and Western blot analysis [224]. Further, amiRNA DENV-128 was able to decrease DENV replication [225]. Additionally, short hairpin RNA (shRNA) and small interfering RNA (siRNA) molecules specific to NS2A demonstrated antiviral properties against JEV in vitro [226,227].

To address the limitations of traditional vaccines, novel vaccine candidates, and genetically engineered flaviviruses with MREs that allow host miRNAs to target viral transcription via endogenous RNA silencing have been proposed. In vivo, the MRE-based WNV vaccine including insertion of several MREs of miR-124a with perfect complementarity in tandem flanking regions of the UTR exhibited safe tolerance and persistent expression. Vaccine construct suppressed WNV infection and produced neutralizing antibody titers that protected animals from lethal challenge [33,229]. This study offers proof in favor of employing miRNAs as an antiviral treatment to combat viruses.

## 7. Exosomes

Exosomes (EVs) or microvesicles released from cells carry viral genomes, proteins, mRNAs, DNA, and miRNAs to target cells, altering the physiological state of the recipient cell. With miRNAs as one of its cargos, EVs can both promote and inhibit viral proliferation; thus, they are the potent players in viral pathogenesis (Table 6) [232,233,234,235]. Viral RNA, DENV-3 290 or DENV-3 5532, was identified in the EVs isolated from DENV-infected human MDDCs. This indicates that DENV utilizes the EV route to differentially modulate immunological response. Tomohiro et al. demonstrated that cell-to-cell transfer of JEV- and DENV-derived sub-genomic replicons contributed to viral propagation and pathogenesis [236].

Of note, EVs derived from PBMCs treated with IFN alpha blocked DENV replication (DENV-3-5532) in recipient cells [237], whereas berbamine treatment decreased the low-density lipoprotein receptor (LDLR) expression at the plasma membrane, which in turn significantly lowered the susceptibility of cells to JEV infection [240]. Interestingly, EVs derived from human saliva showed anti-ZIKV activity [241], while cow- and goat-milk-derived EVs showed anti-DENV activity [245]. Additionally, in EVs overexpressing Defensin alpha 1B (DEFA1), interferon-induced transmembrane protein 3 (IFITM3), and loaded with LL-37, cathelicidin antimicrobial peptide exerts significant anti-ZIKV activity [242,243,244]. Further, enrichment of EVs isolated from cells either infected with WNV and/or stimulated by IFN-alpha revealed differential expression of mRNAs and miRNAs. This suggests that WNV infection modulates the RNA cargo composition in EVs through both IFN-independent and IFN-dependent pathways [238]. Of note, the ZIKV study employing the rhesus macaque trophoblast stem cell model identified EV-derived diagnostic non-invasive miRNA markers that may aid in the diagnosis of placental infection [239]. Unlike other pharmacological molecules, EVs can pass the blood–brain barrier, have low immunogenicity, and are stable in biofluids, rendering it a promising therapeutic option [247,248].

## 8. Clinical Trials Against Flaviviruses

Flaviviruses represent a concern to global health since there are no specific antiviral drugs available for their treatment or prophylaxis. Promising preclinical candidates have been investigated for potential clinical benefits. Clinical trials employing FDA-approved drugs with potent antiviral activity in preclinical testing, such as balapiravir, chloroquine, lovastatin, and celgosivir, have failed to demonstrate efficacy in viral load reduction or beneficial clinical outcomes.

A plethora of antiviral compounds—both direct-acting and host-factor-targeting directed against distinct stages of the viral life cycle—are underway. As per the information retrieved on 1 July 2024 from https://clinicaltrials.gov, Table 7 lists the drugs targeting virion entrance and fusion, NS4B, NS5 polymerase, and host-directed antivirals. The antiviral efficacy of these compounds against JEV, ZIKV, YFV, and DENV is being assessed. Several of these compounds failed in the clinical trial, which thus called for the early conclusion of clinical trials, in addition to trials that have had to conclude early due to a lack of sufficient patients or funding.

## 9. Discussion and Conclusions

Flavivirus infections are a serious global health concern. Direct-acting antivirals with high effectiveness are desperately needed for both preventive and therapeutic action against illnesses caused by mosquito-borne flaviviruses. Different approaches have been implemented to develop prospective antiviral agents against flavivirus infection. These include (i) rational design derived from the crystal structures of viral proteins, (ii) screening the database of FDA-approved synthetic and natural compounds, (iii) testing of identified inhibitors effective against other human viruses, (iv) chemically altering the recognized viral inhibitors to enhance their therapeutic efficacy, (v) intravenous therapy with nucleic acids and immunoglobulins directed against viral genomes, (vi) antiviral peptides from combinatorial libraries by phage display, and (vii) exosomes (EVs) or microvesicles released from virus-infected cells. The implementation of these strategies resulted in the development of some successful antivirals, such as ZIKV-IG (20), ivermectin (251), Omr-IgG-am (NCT00068055), and UV-4B (167), assisting in the treatment of flavivirus infection. However, these approaches do have limitations. For instance, virus-directed inhibitors provide specificity, safety, and enhanced efficacy for treating flavivirus infections (NCT01856205), but this strategy often fails due to the emerging flaviviruses. In this scenario, host-directed inhibitors with potent antiviral activity (251,167) and a high barrier to resistance may be beneficial as they are poorly influenced by viral genetic polymorphisms. Nonetheless, these have caveats because they target host cellular functions that may be critical to cell survival and thus may be toxic. In addition, host-directed inhibitors are vulnerable to host genetic polymorphisms, which may hamper their ability to impair target protein activity.

The successful trial of IVIG provides clinical proof of concept for the development of antiviral treatments (NCT01856205). However, monoclonal antibodies with antiviral activity are also susceptible to mutations in the viral genome, which alters the pathogenic potential of the virus, resulting in the emergence of viral escape mutants, resistant to a specific monoclonal antibody (28). To combat this viral escape phenomenon, certain monoclonal antibodies targeting multiple viral epitopes may be used in conjugation to prevent the neutralization escape by the viruses. Similarly, engineering an RNA silencing mechanism aims to provide antiviral protection. The expression of amiRNAs could be employed to interfere with viral infections with high specificity (196–200). While many amiRNAs have demonstrated high antiviral efficiency, not all amiRNAs directed against viral targets efficiently restrict viral multiplication, leading to the emergence of a broad range of escape mutant viruses. Of note, peptide inhibitors are an appealing alternative as they are less expensive to create and safer than small-molecule- and antibody-based antiviral treatments (9,10,11,15). However, peptides have relatively short half-lives and immunogenicity.

Medicinal plants serve as an excellent source of biodiversity of novel natural compounds for drug discovery pipelines targeting viral infections (32–36). However, there are worries regarding absorption profiles, standardization, uniformity, and stability, in the early stages of drug discovery, to identify natural inhibitors with balanced pharmacokinetics (ADME—absorption, distribution, metabolism, and excretion). In a similar vein, exosomes represent an exciting intervention due to the absence of undesirable factors and their high biocompatibility with living tissues, such as primary human cell cultures and cell lines, as well as also being able to traverse biological barriers, such as the blood–brain barrier (BBB), which enhances bioavailability during neuroinflammation (206–210). Yet, the major considerations for clinically effective exosomes include scalability, repeatability, nontoxicity, property, and potency. Aside from that, the heterogeneity of exosomes, arising from their varied sources and cargo content, are hurdles for consistent therapeutic effectiveness and safety.

The challenges associated with these approaches warrant considerable exploration, necessitating an appropriate animal model showing clinical symptoms. Various animal models for flaviviruses have been tested. A perfect animal model must have reproducible viremia, be immuno-competent, and have viremia for all serotypes; unfortunately, no model meets all of these criteria. To date, mouse models such as AG129 [249] and IFNAR^−/−^ [250] have been employed as a model of ZIKV pathogenesis to evaluate vaccines, therapeutics, and disease pathogenesis. The same mouse model as well as a tree shrew model [251] and A/J mice [252] have been described for DENV pathogenesis. Further, C57BL/6 mice [253], AG129 mice [254], and BALB/c mice [255] have been used as experimental models for JEV infection, whereas AG129 mice [256], IFNAR^−/−^, Syrian golden hamsters [257], and rhesus macaques [258] are primarily used as models for the yellow fever virus. Similarly, immunocompetent mice, hamsters, and geese are susceptible to WN virus infection [259,260,261].

Each of these experimental models resembles human disease to some extent. However, each has advantages and disadvantages when used in preclinical antiviral drug testing. Rodents are not natural flavivirus hosts; hence, they do not develop clinical signs like humans. Hamsters and non-human primates (NHPs) are natural hosts and serve as reservoirs for flaviviruses owing to physiological and genetic similarities with humans. Consequently, they have a benefit over rodents. Nevertheless, one drawback of NHP models is the availability of validated reagents for use with NHP materials.

A mere handful of inhibitors proven effective in animal models have advanced to clinical trials, and the findings cannot be easily extrapolated to humans. This is probably due to the paradoxical involvement of the host immune system in flavivirus infection. As a result, combination therapy trials incorporating both anti-inflammatory and antiviral drugs that have been evaluated in different animal models should be given due consideration.

Research must proceed rapidly, if appropriate, to in vivo investigations and clinical trials. The development of optimal anti-flaviviral drugs may benefit from the re-employment and repurposing of techniques and FDA-approved drugs. A thorough assessment of the drug combination is necessary to identify a beneficial synergy for heightened treatment. A promising method that may be applied to the treatment of flaviviruses is the use of multidrug combinations targeting different stages of viral replication and mitigating the impacts of antiviral resistance.

The insights gleaned from unsuccessful therapy trials may be useful in future research as novel compounds are clinically evaluated for their ability to counter flavivirus infection. The scientific methods for screening and studying inhibitors, which target the crucial host or viral elements involved in the flavivirus life cycle, along with the noteworthy advancements in molecular and structural virology, also offer opportunities and highlights for the advancement of future medical therapies for mosquito-borne flaviviruses. Countermeasures that limit the spread of flaviviruses and disease must be continuously developed and put into action. It is crucial for public health authorities to have surveillance systems that look into and manage the spatial localization of pathogens and their dissemination. This review seeks to highlight the significance of developing novel medications to treat these viruses, which are becoming more widespread worldwide. At present, the majority of nations with a flavivirus epidemic are found in tropical or subtropical regions without having recourse to hospitals or medical personnel. Nevertheless, this reality is about to change due to climate change [262], and even the wealthiest nations will face challenges due to a shortage of antiviral drugs.

## Figures and Tables

**Figure 1 viruses-17-00074-f001:**
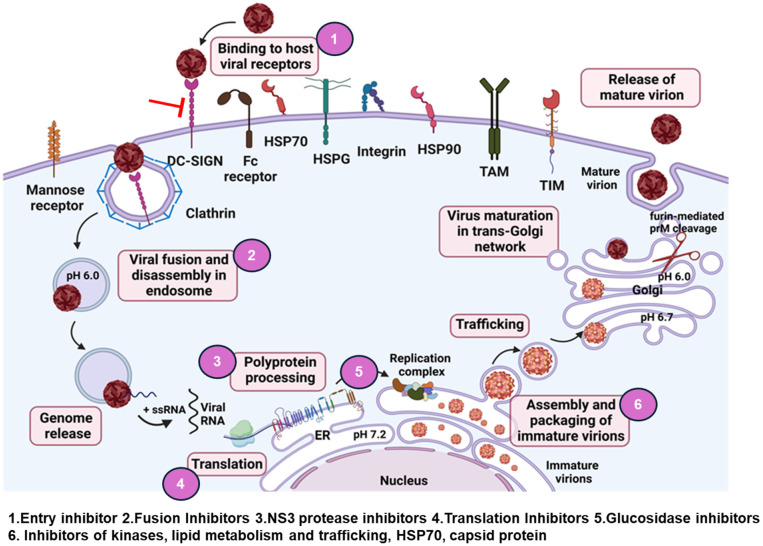
A depiction of the virus life cycle and the inhibitor sites.

**Figure 2 viruses-17-00074-f002:**
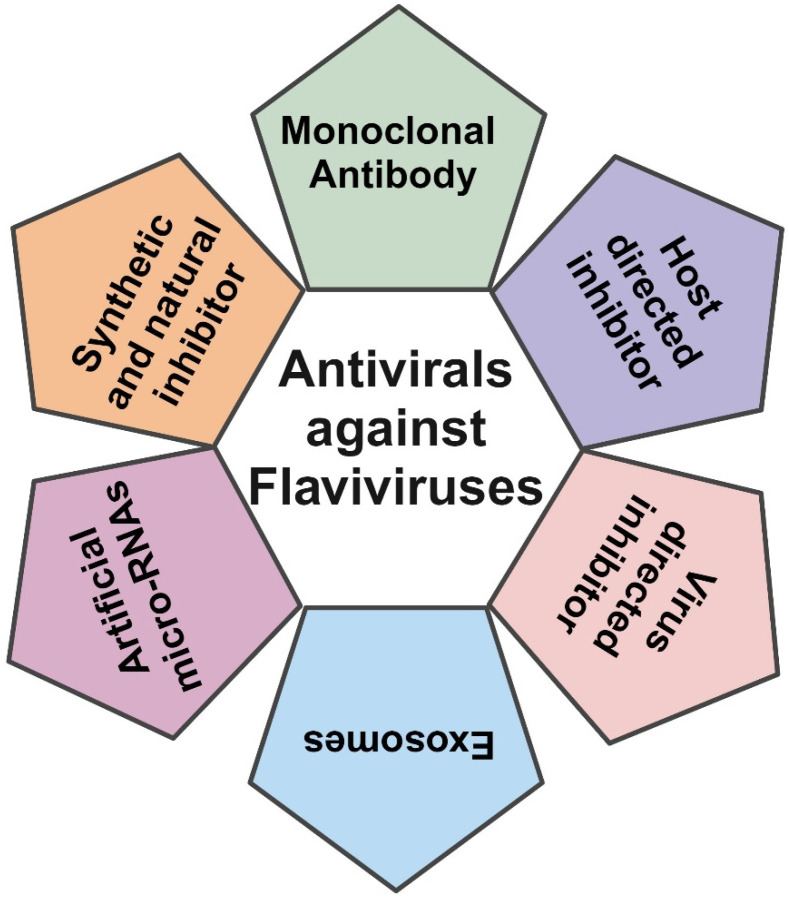
A depiction of the different approaches for developing antivirals against flavivirus.

**Table 1 viruses-17-00074-t001:** Antivirals targeting flavivirus entry.

Target	Drug	Viral Specificity	Study Stage	Ref.
Envelope	Z2	DENV, YFV	In vitro	[5,6]
		ZIKV	In vivo	[5]
	DN59	DENV-2	In vitro	[7,8]
		WNV	In vitro	
	P5	JEV, ZIKV	In vivo	[9]
	DET2 and DET4	DENV-2	In vitro	[10,11]
	Dipeptide EF	DENV	In vitro	[12]
	JBJ-01-162-04	DENV, JEV, WNV, ZIKV	In vitro	[13]
	mAb513	DENV	In vivo	[14]
	2D22	DENV	In vivo	[14,15,16]
	ZIKV-Ig	ZIKV	Phase 1 clinical trial	[17]
	TY014	YFV	Phase 1 clinical trial	[17]
	Tyzivumab	ZIKV	Phase 1 clinical trial	[17]
	DN59	DENV, WNV	In vitro	[18]
	MGAWN1	WNV	Clinical trial withdrawn due to low enrollment	[19]
	3-110-22 (Cyanohydrazone)	DENV, JEV, WNV, ZIKV	In vitro	[20]
	ZKA190 and FIT-1	ZIKV	In vivo	[21]
	ZIKV-117	ZIKV	In vivo	[22]
	EDE1 and EDE2	ZIKV	In vitro	[23]
	mAb WNV-86	WNV	In vivo	[24]
	Gossypol	DENV, ZIKV	In vitro	[25]
	Curcumin	DENV, ZIKV	In vitro	[26,27]
	BP34610	DNEV-1, 2, 3, 4, and JEV	In vitro	[28]
Viral entry	Geraniin	DENV-2	In vivo	[29,30]
	Palmatine	WNV, DENV-2, JEV, YFV, ZIKV	In vitro	[31]
	Prochloroperazine (PCZ)	DENV, JEV	In vitro	[32]
	Daptomycin	JEV	In vitro	[33,34,35]
		Puerto Rico ZIKV		
	Nanchangmycin	CHIKV, DENV, WNV		[34]
	Erlotinib, Sunitinib	DENV	In vivo	[36]
		WNV, ZIKV	In vitro	
	25-Hydroxylcholesterol	DENV, YFV, WNV	In vitro	[37]
		ZIKV	In vivo	
	Chloroquine	ZIKV	In vivo	[38,39]
		DENV	Phase 2 clinical trial failed (no viremia reduction)	[33,40]
	Niclosamide	DENV, WNV, YFV, JEV	In vitro	[41]
		ZIKV	In vivo	
	DET4 and DET2	DENV	In vitro	[10,11]
	EF	DENV	In vitro	[12]
	Ri57	DENV, ZIKV	In vitro	[42]
	AC-10	ZIKV	In vivo	[43]

**Table 2 viruses-17-00074-t002:** Host-directed flavivirus antivirals.

Target	Drug	Viral Specificity	Study Stage	Ref.
Dopamine D2 receptor	Prochloroperazine	DENV, JEV	In vitro	[32]
Clathrin	Chlorpromazine	JEV, WNV	In vitro	[53]
AXL	Nanchangmycin	ZIKV	In vitro	[34]
Lipoxygenase-activating protein	MK-591	ZIKV	In vitro	[35]
Receptor tyrosine kinase and epidermal growth factor tyrosine kinase	Sunitinib and erlotinib	DENV	In vivo	[36]
Adenosine monophosphate-activated protein kinase	PF-06409577, metformin, and AICAR	WNV, ZIKV and DENV	In vitro	[54]
Holesterol-25-hydroxylase	25-hydroxylcholesterol	ZIKV	In vitro	[37]
Coenzyme acetyl	PF-05175157	WNV	In vivo	[55]
Sphingomyelinase-2	GW4869	WNV and ZIKV	In vitro	[56,57]
Endosome acidification	Chloroquine	DENV	In vitro	[58]
Endosomal acidification	Niclosamide	ZIKV	In vitro	[59]
De novo pyrimidine biosynthesis pathway	AR-12, P12-23, and P12-34a	DENV, ZIKV, and JEV	In vitro	[60]

**Table 3 viruses-17-00074-t003:** Antivirals targeting flavivirus replication and polyprotein synthesis and processing.

Target	Drug	Viral Specificity	Study Stage	Ref.
NS1	Peptide 3, 4, 10, 11	DENV	In vitro	[68,69]
	Deoxycalyxin-A	ZIKV	In silico	[70]
	mAb AA12	ZIKV	In vivo	[71]
	mAb 2B7	DENV, WNV, ZIKV	In vitro	[72]
	Castanospermine and deoxynojirimycin	DENV, ZIKV	In vitro	[73]
	N-nonyl-Deoxynojirimycin	DENV-2, JEV	In vitro	[74]
	Celgosivir	DENV	Phase 1 clinical trial	[75,76]
NS2A	Novobiocin	DENV, ZIKV	In vivo	[77,78]
NS2B–NS3	Temoporfin	DENV, YFV, WNV, JEV	In vitro	[65]
		ZIKV	In vivo	[65]
	JMX0207	DENV-2	In vitro	[79]
		ZIKV	In vivo	[77]
	Nelfinavir	DENV-2	In vitro	[80]
	Compound 4	ZIKV	In vitro	[81]
	Compound 14, Compound 15	DENV	In vitro	[82]
	NSC135618	DENV, ZIKV, WNV, YFV	In vitro	[83]
	ZP10	ZIKV	In vitro	[84]
	*MB21*	DENV	In vitro	[85]
	Aprotinin	DENV	In vitro	[85,86]
		WNV	In silico	
	Hydroxychloroquine	ZIKV	In vivo	[87,88]
	Methylene blue	ZIKV	In vivo	[89]
		DENV	In vitro	[79,90,91]
	Erythrosin B	ZIKV, DENV	In vitro	
	Ivermectin	YFV, WNV	In vitro	[92]
		ZIKV	In vivo	[93]
		DENV	Phase 2/3 clinical trial	[94]
	Myricetin, quercetin, luteolin, isorhamnetin, apigenin, and curcumin	ZIKV	In vitro	[51]
	Bortezomib	ZIKV, DENV	In vitro	[95]
	Palmatine	DENV, JEV, WNV, ZIKV	In vitro	[31]
	Asunaprevir, Simeprevir	ZIKV	In vitro	[96]
NS3	ST-610	DENV	In vivo	[97]
	SYC-1307	DENV, JEV, WNV, ZIKV	In vivo	[98]
	ML283	DENV, WNV	In vivo	[99]
	Suramin	DENV, ZIKV	In vitro	[100]
NS2B	NSC157058	WNV	In silico	[101]
		ZIKV	In vivo	
NS4A	Compound B and SBI-0090799	DENV, ZIKV	In vitro	[102]
NS3-NS4B	JNJ-A07	DENV	In vivo	[103]
	JNJ-64281802	DENV	Phase 2 clinical trial	
	JNJ-1802	DENV	Phase 1 clinical trial	[104]
		JEV, WNV, ZIKV	In vitro	
NS4B	NITD-688	DENV	In vivo	[105]
	Manidipine	JEV	In vivo	[106]
		DENV, ZIKV, WNV	In vitro	
	JMX0254	DENV, YFV	In vivo	[107,108]
	SBI-0090799	ZIKV	In vitro	[109]
	AZD0530, dasatinib	DENV-2	In vitro	[110,111]
NS5	Galidesivir	WNV, TBEV, ZIKV	In vivo	[112,113]
		YFV	Phase I clinical trial	[114]
	Favipiravir	WNV, YFV	In vitro	[115]
		ZIKV	In vivo	[116]
	Balapiravir	DENV	Phase 1/2 clinical trial	[117]
	NITD-008	ZIKV, TBEV, DENV	In vivo	[118,119]
	AT-752	DENV, YFV	Phase II clinical trial	[120,121]
		WNV, ZIKV, JEV	In vitro	
	7DMA	TBEV, ZIKV, WNV, DENV	In vivo	[111,122,123]
	Sofosbuvir	YFV	In vivo	[124]
		ZIKV	In vivo	[125,126]
	NITD-434, NITD-640	Pan-flavivirus	In vitro	[127]
	NITD-29	DENV	In vitro	[128]
	Efavirenz, tipranavir, dasabuvir	WNV, ZIKV, TBEV	In vitro	[129]
	AR-12	DENV	In vivo	[60]
	P12-23, P12-34	DENV, ZIKV, JEV	In vitro	
	Ribavirin	ZIKV	In vivo	[130]
	Ivermectin	ZIKV	In vitro	[131]
		DENV	Phase 2/3 clinical trial	[132]
	T-1106	YFV	In vivo	[133]
	Rifapentine	YFV	In vivo	[6]
	Emetine	ZIKV	In vivo	[134]
	Lycorine	ZIKV	In vivo	[135]
	Dolutegravir	ZIKV	In vitro	[136]
	Compound TPB	ZIKV	In vivo	[137]
	Sinefungin	WNV, DENV	In vivo	[138,139,140]
	Chartaceones	ZIKV	In vitro	[52]
	Fleximers	DENV, ZIKV, YFV	In vitro	[141,142]

**Table 4 viruses-17-00074-t004:** Antivirals targeting flavivirus assembly and egress.

Target	Drug	Viral Specificity	Study Stage	Ref.
Assembly	Deoxynojirimycin (DNJ)	DENV	In vivo	[74]
	Castanospermine	DENV	In vivo	[187]
	Celgosivir	DENV	Phase 1/2	[188,189,199]
	UV-4B	DENV	Phase 1/2	[190]
	UV-12	DENV	In vivo	[191,192,193]
	CM-9-78			
	CM-10-18			
	PF-05175157	WNV, DENV, ZIKV	In vivo	[55]
	TOFA		In vitro	[55,194]
	MEDICA 16		In vitro	
	SFV785	DENV, YFV	In vitro	[195,196]
	Lovastatine	DENV	In vivo	[197,198,200]
	Atorvastatine, ezetimibe	DENV, ZIKV	In vitro	[201]
Capsid	VGTI-A3	DENV	In vitro	[194]
	VGTI-A3-03		In vitro	[202]
	ST-148		In vivo	[203,204]

**Table 5 viruses-17-00074-t005:** This table lists publications that implicate artificial/mi/si/sh RNAs as having potential against flaviviruses.

Artificial/mi/si/sh RNAs	Viral Specificity	Study Stage	Ref.
amiRNA #1 and amiRNA #2	JEV	In vitro	[223]
amiRNA #1 and amiRNA #2	WNV	In vitro	[224]
amiRNA DENV-128	DENV	In vitro	[225]
miR-124a	WNV	In vivo	[228]
miR-155	WNV	In vivo	[229]
shRNA	YFV	In vivo	[230]
siRNA	ZIKV	In vivo	[231]

**Table 6 viruses-17-00074-t006:** This table lists publications that implicate exosomes as having potential against flaviviruses.

Viral Specificity	Study Stage	Ref.
DENV	In vitro	[237]
WNV	In vivo	[238]
ZIKV	In vitro	[239]
JEV	In vivo	[240]
ZIKV	In vivo	[241]
ZIKV	In vivo	[242]
ZIKV	In vivo	[243]
ZIKV	In vivo	[244]
ZIKV	In vitro	[241]
DENV	In vitro	[245]
ZIKV	In vivo	[246]

**Table 7 viruses-17-00074-t007:** Clinical trials on drugs targeting flaviviruses.

Antiviral	Target	Virus	Clinical Trial Identifier	Clinical Trial	Status	Note	Ref.
TY014	Envelope	YFV	NCT03776786	Phase 1	Completed	Safe and abrogate viremia	Clinical trials.Gov (accessed on 1 July 2024); available online: Study Details/Safety and Tolerability of an Antibody Against Yellow Fever Virus (TY014) in Humans/ClinicalTrials.gov/NCT03776786
Zika virus immune globulin (ZIKV-IG)	Envelope	ZIKV	NCT03624946	Phase 1	Completed	Safe and well tolerated	Clinical trials.Gov (accessed on 1 July 2024); available online: Study Details/Study in Healthy Volunteers Evaluating Safety and Pharmacokinetics of Zika Virus Immune Globulin (ZIKV-IG)/ClinicalTrials.gov/NCT03624946
Tyzivumab	Envelope	ZIKV	NCT03443830	Phase 1	Completed	No results available	Clinical trials.Gov (accessed on 1 July 2024); available online: Study Details/Safety and Tolerability of an Antibody Against Zika Virus (Tyzivumab) in Humans/ClinicalTrials.gov/NCT03443830
Intravenous immunoglobulin (IVIG)	Envelope	JEV	NCT01856205	Phase 2	Completed	JEV patients had a greater increase in neutralizing antibody titers	Clinical trials.Gov (accessed on 1 July 2024); available online: Study Details/Safety and Efficacy Study of Intravenous Immunoglobulin to Treat Japanese Encephalitis/ClinicalTrials.gov/NCT01856205
Omr-IgG-am	Envelope	WNV	NCT00068055	Phase 1/2	Completed	Safe and well-tolerated in patients	Clinical trials.Gov (accessed on 1 July 2024); available online: Study Details/IVIG—West Nile Encephalitis: Safety and Efficacy/ClinicalTrials.gov/NCT00068055
MGAWN1	Envelope	WNV	NCT00515385	Phase 1	Completed	Safe in healthy subjects	Clinical trials.Gov (accessed on 1 July 2024); available online: Study Details/A Trial to Evaluate the Safety of a Single Intravenous Infusion of MGAWN1 in Healthy Adults/ClinicalTrials.gov/NCT00515385
Carica papaya leaf extract (CPLE)	NS2B–NS3	DENV	NCT06121934	Phase 3	Completed	No results available	Clinical trials.Gov (accessed on 1 July 2024); available online: Study Details/Efficacy and Safety of Carica Papaya in Dengue Fever: A Randomised Clinical Trial//ClinicalTrials.gov/NCT06121934
Ribavirin	NS3 helicase	JEV	NCT00216268	Phase 2	Unknown	No results available	Clinical trials.Gov (accessed on 1 July 2024); available online: Study Details/Treatment of Japanese Encephalitis/ClinicalTrials.gov/NCT00216268
EYU688	NS4B	DENV	NCT06006559	Phase 2	Ongoing	No results available	Clinical trials.Gov (accessed on 1 July 2024); available online: Study Details/A Study to Assess the Efficacy, Safety and Pharmacokinetics of EYU688 in Patients With Dengue Fever/ClinicalTrials.gov/NCT06006559
Galidesivir	NS5	YFV	NCT03891420	Phase 1	TERMINATED	Withdrawn due to lack of funding	Clinical trials.Gov (accessed on 1 July 2024); available online: Study Details/A Study to Evaluate the Safety, Pharmacokinetics and Antiviral Effects of Galidesivir in Yellow Fever or COVID-19/ClinicalTrials.gov/NCT03891420
Iron-fortified food	Oxidative stress-mediated signaling	JEV	NCT06027801	Not Applicable	Recruiting	No results available	Clinical trials.Gov (accessed on 1 July 2024); available online: Study Details/Iron Fortified Food to Improve Japanese Encephalitis and Typhoid Fever Vaccine Immunogenicity/ClinicalTrials.gov/NCT06027801
Metformin	AMPK	YFV	NCT04267809	Phase 2	Completed	No results available	Clinical trials.Gov (accessed on 1 July 2024); available online: Study Details/Modulate Cellular Stress in the Immune Cells to Reduce Rate of Symptomatic Viral Infection/ClinicalTrials.gov/NCT04267809
AVI-4020 injection	Interfere with RNA–RNA duplex structures	WNV	NCT00387283	Phase 1	Completed	No results available	Clinical trials.Gov (accessed on 1 July 2024); available online: Study Details/Pharmacokinetic Study in Cerebral Spinal Fluid After a Single Dose of AVI-4020/ClinicalTrials.gov/NCT00387283
Ivermectin	IMPα/β (Host)	DENV	NCT02045069	Phase 2/3	Unknown	No results available	Clinical trials.Gov (accessed on 1 July 2024); available online: Study Details/Efficacy and Safety of Ivermectin Against Dengue Infection/ClinicalTrials.gov/NCT02045069
Ivermectin	IMPα/β (Host)	DENV	NCT03432442	Phase 2	Completed	Safe and accelerated NS1 antigen clearance in dengue patients	Clinical trials.Gov (accessed on 1 July 2024); available online: Study Details/Pharmacokinetics and Pharmacodynamics of Ivermectin in Pediatric Dengue Patients/ClinicalTrials.gov/NCT03432442
UV-4B	ER α-glucosidase I and II	DENV	NCT02061358	Phase 1	Completed	Safe in healthy subjects	Clinical trials.Gov (accessed on 1 July 2024); available online: Study Details/Study to Determine the Safety, Tolerability and Pharmacokinetics of UV-4B Solution Administered Orally in Healthy Subjects/ClinicalTrials.gov/NCT02061358
Chloroquine	Endosomal acidification	DENV, ZIKV	NCT00849602	Phase 1/2	Completed	CQ ameliorated pain and lowered the viral load in dengue patients	Clinical trials.Gov (accessed on 1 July 2024); available online: Study Details/The Effect of Chloroquine in the Treatment of Patients With Dengue/ClinicalTrials.gov/NCT00849602
Dexamethasone	NF-kB and activator factor-1	DENV	NCT05631405	Not Applicable	Ongoing	No results available	Clinical trials.Gov (accessed on 1 July 2024); available online: Study Details/Efficacy of Dengue Infection With Warning Signs Treated With Dexamethasone (DengDex Study)/ClinicalTrials.gov/NCT05631405
Anti-d	Fcγ receptor	DENV	NCT01443247	Not Applicable	Completed	No results available	Clinical trials.Gov (accessed on 1 July 2024); available online: Study Details/Role of Andi-d in Dengue Fever: a Pilot Study/ClinicalTrials.gov/NCT01443247
Anakinra	Interleukin-1 receptor (IL-1R)	DENV	NCT05611710	Phase 2	Ongoing	No results available	Clinical trials.Gov (accessed on 1 July 2024); available online: Study Details/Anakinra in Dengue With Hyperinflammation (AnaDen)/ClinicalTrials.gov/NCT05611710

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
