# Peer review of "A Comprehensive Review of the Development and Therapeutic Use of Antivirals in Flavivirus Infection"

_viruses, 2025, doi:10.3390/v17010074_

Round 1

Reviewer 1 Report

Comments and Suggestions for Authors

Broaden the discussion to include challenges associated with antiviral development. This could involve emerging resistance, limitations of animal models, and discrepancies between in vitro and in vivo results. Compare the compounds tested with successful antivirals in other viral families to highlight lessons learned.

Incorporate a comparative analysis (advantages/disadvantages) of the strategies presented. For example, assess whether inhibitors targeting host proteins have more significant clinical potential than those targeting viral proteins, considering the risk of side effects or the ability to circumvent viral resistance.

Include relevant quantitative data, such as percentages of effectiveness in preclinical trials and IC50s of crucial inhibitors.

Author Response

Reviewer 1

Broaden the discussion to include challenges associated with antiviral development. This could involve emerging resistance, limitations of animal models, and discrepancies between in vitro and in vivo results. Compare the compounds tested with successful antivirals in other viral families to highlight lessons learned.

Incorporate a comparative analysis (advantages/disadvantages) of the strategies presented. For example, assess whether inhibitors targeting host proteins have more significant clinical potential than those targeting viral proteins, considering the risk of side effects or the ability to circumvent viral resistance.

Include relevant quantitative data, such as percentages of effectiveness in preclinical trials and IC50s of crucial inhibitors.

Response: We would like to thank the reviewer for the insightful comments that will help in improving the manuscript quality further. We agree with the reviewer to add the percentage of effectiveness in preclinical trials and IC50 value, hence we have updated the IC50 and EC50 value throughout the manuscript. We have also broadened the discussion part and tried to add as much information as includes challenges associated with antiviral development and limitations of animal models. We updated the discussion part from line number 797 to 863.

Reviewer 2 Report

Comments and Suggestions for Authors

Flavivirus infections e.g., WNV, ZIKA, DENV, JEV are serious global health concerns with no licensed antiviral drug existing to treat these diseases. This manuscript “Comprehensive review on the development and therapeutic use of antivirals in flavivirus infection” presents a valuable and thorough review of the current efforts and advancements in the development of antiviral therapies for flavivirus infections. The authors effectively present a comprehensive review regarding the biology of flaviviruses, their immune evasion mechanisms, and the progress made in targeting these viruses at different steps and aspects of their life cycle. This review is well-researched, timely, and provides a critical resource for researchers and clinicians in the field.

The paper provides an in-depth analysis of a wide range of antiviral strategies, from viral entry inhibitors to host-directed therapies to different antiviral therapeutics against various viral proteins, which enhances its utility as a reference for the field. The content in this manuscript is clearly laid out with each section focusing on specific stages of the virus life cycle or antiviral targets.

Some minor areas could be improved to make this paper more accessible to readers.

1. Include a flavivirus lift cycle diagram to guide the audience on different sections

2. Add a more detailed discussion to evaluate the limitations of certain drug classes. The paper could benefit from adding a more balanced discussion of failed or withdrawn trials would provide valuable lessons for future research.

Author Response

Reviewer 2

Flavivirus infections e.g., WNV, ZIKA, DENV, JEV are serious global health concerns with no licensed antiviral drug existing to treat these diseases. This manuscript “Comprehensive review on the development and therapeutic use of antivirals in flavivirus infection” presents a valuable and thorough review of the current efforts and advancements in the development of antiviral therapies for flavivirus infections. The authors effectively present a comprehensive review regarding the biology of flaviviruses, their immune evasion mechanisms, and the progress made in targeting these viruses at different steps and aspects of their life cycle. This review is well-researched, timely, and provides a critical resource for researchers and clinicians in the field.

The paper provides an in-depth analysis of a wide range of antiviral strategies, from viral entry inhibitors to host-directed therapies to different antiviral therapeutics against various viral proteins, which enhances its utility as a reference for the field. The content in this manuscript is clearly laid out with each section focusing on specific stages of the virus life cycle or antiviral targets.

Some minor areas could be improved to make this paper more accessible to readers.

Include a flavivirus lift cycle diagram to guide the audience on different sections

Add a more detailed discussion to evaluate the limitations of certain drug classes. The paper could benefit from adding a more balanced discussion of failed or withdrawn trials would provide valuable lessons for future research.

Response: We would like to thank the reviewer for appreciating the concept, writing and giving insightful comments for further improvement of the manuscript. As per the reviewer suggestion, we have included the life cycle of the flavivirus including attachment, entry of the virus, replication, assembly, and finally egress. We have also highlighted the stages of life cycle where the inhibitors act as antiviral. We added the limitations of drugs in the discussion part. Please refer to line # 797 to 863.
